# Protein Hydrolysates from Flaxseed Oil Cake as a Media Supplement in CHO Cell Culture

Marijan Logarušić [ID], Višnja Gaurina Srček, Sara Berljavac, Andreja Leboš Pavunc, Kristina Radošević [ID] and Igor Slivac *[ID]

Faculty of Food Technology and Biotechnology, University of Zagreb, Pierottijeva 6, 10000 Zagreb, Croatia; mlogarusic@pbf.unizg.hr (M.L.); vgaurinasrcek@pbf.unizg.hr (V.G.S.); sberljavac@gmail.com (S.B.); alebos@pbf.unizg.hr (A.L.P.); krado@pbf.unizg.hr (K.R.)

* Correspondence: islivac@pbf.unizg.hr

**Abstract:** This is the first report about flaxseed protein hydrolysates applied as media supplements in CHO cell culture. The hydrolysates were produced by three separate enzymatic digestions of proteins isolated from flaxseed oil cake. The enzymes used were *Alcalase*, *Neutrase*, and *Protamex*, and the most efficient hydrolysis was achieved with *Alcalase*. The three hydrolysates were first tested as a partial substitute for serum in basal media in order to evaluate their effects on the adherent IgG-producing CHO cell line. The cells that grew in such media reached higher density than the cells in media supplemented with serum only. Consequently, the increased cell number improved the final IgG titer. In the next experiment, the impact of hydrolysates was evaluated in suspension CHO culture adapted to chemically defined media. In this preliminary investigation, the cells showed no response to the hydrolysate addition concerning the growth rate and productivity. Despite this outcome, we speculate that low molecular mass components in the hydrolysates, besides nutritive, may have a cell-protective function.

**Keywords:** flaxseed oil cake; protein hydrolysates; cell culture media supplementation; CHO cell technology



## 1. Introduction

Animal cells are a complex expression system applied in a wide range of life science research as well as in industrial production of various biologicals. The growth-supporting microenvironment of cells is largely defined by nutrient composition of cell culture media. Commonly used media formulations are buffered solution of salts, sugars, vitamins, and amino acids (e.g., Dulbecco's Modified Eagle's Medium, DMEM). Proteins and peptides are essential for cell proliferation as the source of building blocks and energy as well as signaling molecules, thus they have a key role in media preparation. The standard supply of proteins and peptides in media is serum (5–20% *v*/*v*), a complex mixture of macromolecules derived from blood of fetal calves (FBS—fetal bovine serum). Due to its biological source, serum has undefined composition, which can lead to extended downstream processing and result in batch-to-batch variability, and even contamination with blood-borne pathogens. This made many regulatory authorities recommend avoiding ingredients of animal origin in bioprocessing production as much as possible. Consequently, more and more laboratories and bioprocessing companies turn to chemically defined media (CDM) composed only of traceable and non-animal-derived components. Although the CDM market has increased in the last 15 years, these media are still relatively expensive since they often have to be suitably designed. Moreover, the cell adaptation to serum-free conditions is difficult, time-consuming, and stressful for the cells, and thus not always successful [1]. Alternatively, an adjustment can be made by application of plant-derived protein hydrolysates. These kinds of hydrolysates have been used occasionally for almost five decades for the reduction or replacement of FBS from basal media, such as DMEM [2,3]

or enrichment of CDM [4]. They are produced by enzymatic or acidic digestion of a given plant material, very often originating form protein-rich substrates such as seeds of soy, rapeseed, sunflower, cotton, palm, sesame, flax, etc. [5,6]. The product of digestion consists of peptides, free amino acids, residual oils, minerals, and some unidentified components of unknown biological activity. Therefore, it is necessary to experimentally determine the proper hydrolysate dosage for a given medium formulation that provides the desired effects, such as improved cell growth, enhanced cell viability, and increased recombinant protein production. CHO cells (Chinese hamster ovary cells) have been the primary platform for production of recombinant glycoproteins for more than 30 years. The majority of high-value therapeutic proteins today are produced with CHO cells. They are quite robust and can grow as adherent or suspension cell culture, in serum-containing media or CDM. Considerable efforts have been devoted to the advancement of their performance by optimizing bioprocess conditions, including culture media formulation [7,8]. Supplying the media with alternative peptide sources, such as the plant protein hydrolysates, has also been a part of the optimization strategy [4,9]. In this work, we describe adherent CHO DP-12 cell growth and monoclonal antibody (mAb) production in DMEM, where serum is partially replaced by flaxseed protein hydrolysates. A brief screening of the suspension of CHO cells in CDM supplemented with the hydrolysates was also conducted. The hydrolysates were produced in-house, with three separate digestions using different enzymes. The source of the proteins for digestion was flaxseed oil cake, a crude residue obtained after oil extraction from flaxseeds. Oil cakes are a regular leftover product of edible oil manufacturing and their utilization largely depends on their origin. Since flaxseed oil cakes are very rich in valuable nutrients, especially proteins (20–30%) and leftover fibers and lipids [10], the sustainability enthusiasts find various ways for their exploitation [11–13]. Proteins from flaxseed oil cake are rich in arginine and glutamate as well as the branched-chain amino acids such as lysine, leucine, and valine. Besides being essential exogenous metabolic intermediates, these amino acids exhibit free radical scavenging activity and inhibit linoleic acid oxidation [14]. The amino acid composition of the flaxseed protein isolate shares considerable similarity to whey and soy protein isolates, which have also been used as cell media supplements in hydrolysate form [15,16]. Additionally, peptides and peptide fractions obtained from flaxseed proteins show high antioxidant activity [17], which can be beneficial for in vitro growing cells, as shown in [18] for murine macrophages and in [19] for HeLa cells. Digestion of the proteins in this study was carried out using three commercial endopeptidases: *Alcalase*$^{TM}$, *Neutrase*$^{TM}$, and *Protamex*$^{TM}$. All three enzymes are of microbial origin and classified as food-grade, but they have different specificities. Exopeptidases were not chosen in order to avoid the production of free amino acids. Improved CHO cell growth and/or productivity were so far reported for commercial or in-house produced protein hydrolysates made from soy [20–22], wheat [21–23], cotton [22], rice [23,24], and rapeseeds [25]. CHO cells were applied for the assessment of antioxidant potential of perilla seeds' protein hydrolysate [26] as well as for chemometric analysis of metabolized soy protein hydrolysate [27]. To our knowledge, this work is the first report about enzymatically digested flaxseed proteins used as media supplements for CHO cell culture.

## 2. Materials and Methods

In Figure 1 the scheme of our experimental setup is illustrated, showing the main steps in production and cell culture assessment of flaxseed protein hydrolysates.

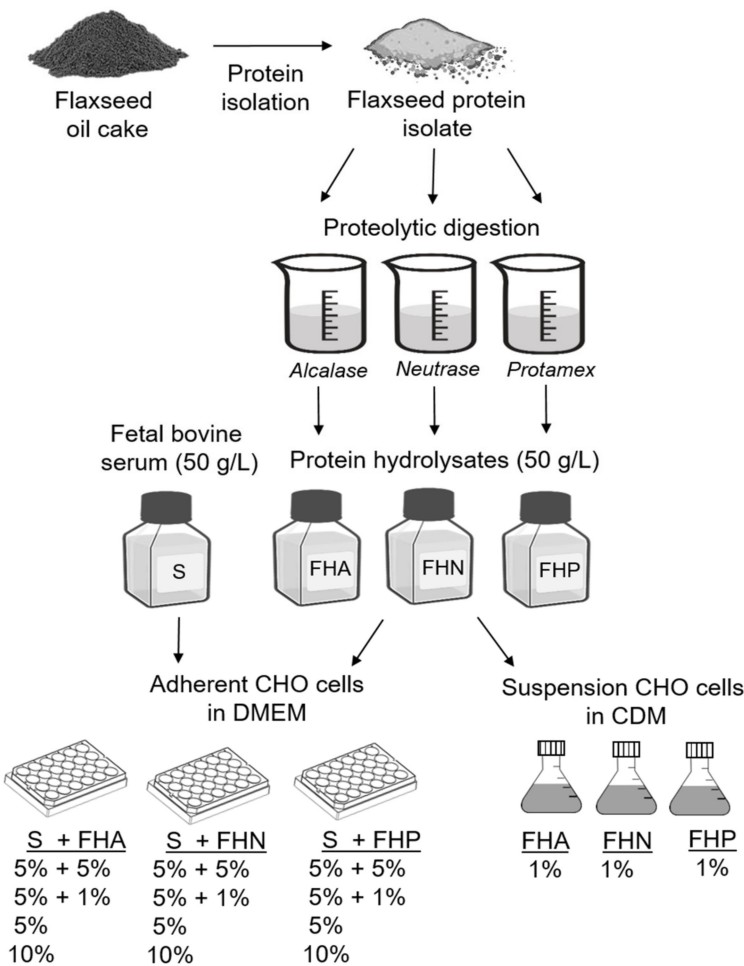

**Figure 1.** Preparation of protein hydrolysates from flaxseed oil cake and their assessment as growth media supplements in CHO cell culture (DMEM—basal media, CDM—chemically defined media, S—fetal bovine serum, FHA/N/P—flaxseed protein hydrolysates made with *Alcalase/Neutrase/Protamex*).

## 2.1. Materials

Flaxseed meal was obtained from a local producer (OPG Jankovic, Zagreb, Croatia). *Alcalase* (2.4 Ug$^{-1}$), *Neutrase* (0.8 Ug$^{-1}$), *Protamex* (1.5 AU-Ng$^{-1}$), *Cellulase* (0.8 Umg$^{-1}$), insulin solution human, Trace Elements A, Trace Elements B, alanyl-glutamine solution, and antibiotic-antimycotic solution were purchased from Sigma-Aldrich (Taufkirchen, Germany). Dulbecco's Modified Eagle's Medium (DMEM) and fetal bovine serum (FBS) were purchased from Capricorn Scientific GmbH (Ebsdorfergrund, Germany). Methotrexate (MTX) was purchased from Cerilliant (Round Rock, TX, USA). PowerCHO®-2 CD chemically defined selective medium was purchased from Lonza (Verviers, Belgium). Anticlumping agent was purchased from GIBCO by Life Technologies (Paisley, UK). All other chemicals and reagents used in this study were of analytical grade.

## 2.2. Preparation of Flaxseed Protein Isolates and Hydrolysates

The proteins to be hydrolyzed were isolated from the flaxseed oil cake produced by a local farmer and edible oil manufacturer (central Croatia, harvest 2019). The proteins were extracted according to the method described in [18] with minor modifications. The defatted flaxseed meal was mixed with 20-fold (*w/v*) deionized water at 37 °C and the mixture was adjusted to pH 5 with 2 M HCl. The *Cellulase* (1%, *w/w*) was added in the mixture to start fiber hydrolysis. After 4 h of reaction, pH of the mixture was adjusted to 10 with 2 M

NaOH in order to achieve *Cellulase* inhibition and alkaline solubilization. The suspension was centrifuged ($8000\times g$, 30 min, 10 °C) and the supernatant was adjusted to pH 4.2 with 2 M HCl. Protein precipitate was collected by mixture centrifugation ($8000\times g$, 30 min, 10 °C) and washed with acidified water (pH 4.2) and centrifuged ($8000\times g$, 30 min, 10 °C) two times. The obtained protein precipitate was resuspended in deionized water, and after homogenization, the suspension was set to pH 7.0 with 2 M NaOH. Before lyophilization, the suspension was dialyzed with SnakeSkin™ Dialysis Tubing, 10K MWCO (Thermo Scientific, Waltham, MA, USA) at 4 °C overnight. The obtained flaxseed protein isolate (FPI) (5%, *w/v*) was dispersed in deionized water. The suspension was heated to optimal catalytic temperatures of individual microbial proteases, prior to adjusting the pH of the suspension to appropriate values with 2 M NaOH or HCl. The hydrolysis was carried out using the following microbial enzymes, at an enzyme–substrate ratio of 1:20, under their optimal reaction conditions: *Alcalase* (A) (pH 8.5, 55 °C), *Neutrase* (N) (pH 7, 55 °C), and *Protamex* (P) (pH 7.7, 50 °C). The temperature and pH of the mixtures were kept constant during hydrolysis. After 4 h of the hydrolysis process, the enzyme reaction was stopped by heat treatment in boiling water for 10 min. The pH of the cooled enzyme digests was adjusted to 7 with 2 M NaOH or HCl. The digest was centrifuged at $5000\times g$ for 20 min to remove insoluble residues while supernatant, containing desired peptides, was collected. Three flaxseed protein hydrolysates (FHA, FHN, and FHP) were sterile filtered (0.22 μm), and their total peptide content was quantified by the Lowry assay. The final hydrolysate concentration was set to 50 g/mL with sterile redistilled water. The hydrolysates were stored at −20 °C until further use. The degree of hydrolysis (DH) was determined according to the method reported in [28]. Briefly, the aliquots of the enzyme digest were taken after 240 min of hydrolysis, and were mixed with the equivalent volume of 0.44 M trichloroacetic acid (TCA). The mixtures were kept at room temperature for 30 min and centrifuged at $10,000\times g$ for 10 min. Protein content of TCA-soluble peptide fraction and hydrolysate sample without TCA was determined by the Lowry assay. DH was expressed as the ratio of soluble protein quantity in 0.22 M TCA to the total protein quantity. SDS-PAGE was performed using precast 4–20% Mini-PROTEAN® TGX gel (BIO-RAD, Hercules, CA, USA) and Mini-PROTEAN 2D Electrophoresis Cell (BIO-RAD, Hercules, CA, USA). Aliquots of protein hydrolysates were diluted 10x with redistilled water and mixed with 5x Laemmli buffer (10% (*w/v*) SDS, 40% (*v/v*) glycerol, 0.01% (*w/v*) bromophenol blue, 25% (*v/v*) β-mercaptoethanol, and 250 mM Tris-HCl, pH 6.8). The samples were boiled for 3 min before loading the gel. Each loaded sample contained 50 μg of peptides. The electrophoresis was performed at 180 V for 50 min. The gel was stained with 0.2% (*w/v*) Coomassie brilliant blue in 50% (*v/v*) methanol and 7% (*v/v*) acetic acid during 1 h and unstained in 7% (*v/v*) acetic acid.

*2.3. Adherent Cell Culture and Media*

CHO DP-12 clone #1934 (ATCC CRL12445) (LGC standards, Teddington, UK) is a cell line that produces a recombinant human monoclonal antibody anti-IL-8, isotype IgG1. The cell propagation of adherent CHO DP-12 cells was performed in T-75 flasks in basal DMEM medium supplemented with 5% fetal bovine serum (FBS), 200 nM methotrexate (MTX), 0.002 mg/mL insulin solution human, 0.1% (*v/v*) Trace Element A, 0.1% (*v/v*) Trace Element B, and 1% (*v/v*) antibiotic-antimycotic solution. CHO DP-12 cells were incubated at 37 °C in a humidified atmosphere containing 5% $CO_2$, and passaged every 4 days. To investigate the hydrolysates, $10^4$ cells/well were seeded in 24-well plates. The media used was DMEM supplemented with 5% (*v/v*) FBS, and 5% or 1% (*v/v*) of one of the three hydrolysates (HA, HN, and HP). The complete DMEM with 5% or 10% FBS, without hydrolysates, were the control media. All the media preparations were tested in triplicates. Since the total protein content in FBS, HA, HN, and HP was around 50 g/L (determined by the Lowry assay), the addition of these supplements equally affected dilution of DMEM components. The cells were kept in the given conditions until their number started to decrease.

### 2.4. Suspension Cell Culture and Media

For the new set of experiments, the CHO DP-12 cell line was adapted to growth in suspension and in serum-free media (data not shown). The new CHO DP-12 strain (CHO DP-12-S) was maintained in PowerCHO®-2 CD serum-free medium in 125 mL Erlenmeyer shake flasks (Corning, Corning, NY, USA) on an orbital shaker (Biosan, Riga, Latvia) at 160 rpm. To make the media complete, the following components were added: 200 nM MTX, 0.002 mg/mL human insulin solution, 1% (*v/v*) antibiotic-antimycotic solution, 4% (*v/v*) alanyl-glutamine solution, and 0.25% (*v/v*) anti-clumping agent. The shake flasks were incubated at 37 °C in a humidified atmosphere containing 5% $CO_2$, and passaged when the cells' concentration reached around $2 \times 10^6$ cells/mL. For the assessment of flaxseed protein hydrolysates, the media were supplied with HA, HN, and HP 1% (*v/v*). Medium without hydrolysates served as a control. Each condition was performed in triplicate (i.e., three flasks per condition). The initial cell concentration in the flasks was $2.5 \times 10^5$ cells/mL in 25 mL of media. The flasks were kept under constant shaking inside the incubator until the cells reached the early decline phase.

### 2.5. Cell Growth, Metabolite, and Recombinant IgG Determination

The cell culture samples were taken daily. Adherent cells were detached by Trypsin EDTA (0.25%) and suspended in fresh DMEM, after the culture media was removed from the wells. The suspension cell culture samples were taken directly from the shake flasks (0.25 mL). Total cell density and viable cell density were determined by the trypan blue exclusion method using a hemocytometer. Aliquots of spent media, after mild centrifugation, were taken and kept frozen at $-20$ °C for further analysis. The concentrations of glucose and lactate in media were determined following the manufacturer's instructions for enzyme-based kits: Glucose GOD-PAP assay (BIOLABO, Maizy, France) and Fluitest® Lactate assay (Analyticon, Lichtenfels, Germany), and using a spectrophotometer (Thermo Scientific Genesys 10S). The concentration of the recombinant IgG in media collected on the final cultivation day was determined by the Turbitex® IgG-2 assay (Analyticon, Lichtenfels, Germany) in accordance with the manufacturer's instructions and was expressed as relative volumetric productivity. The applied IgG assay was based on immunological agglutination. Anti-IgG antibodies react with antigen in the sample to form an antigen/antibody complex. The level of agglutination, i.e., turbidity, was measured by spectrophotometer at 340 nm.

### 2.6. Statistical Analysis

The statistical analysis was performed using the Statistica software (StatSoft, Inc., Tulsa, Ok, USA, 2014) version 12. All experiments were performed in triplicate, and the results are presented as the means $\pm$ SD of the control cells. One-way analysis of variance (ANOVA), followed by Tukey's HSD test, was used to determine statistically significant ($p < 0.05$) differences from the control cells.

## 3. Results and Discussion

The goal of this work was to assess flaxseed protein hydrolysates as additives in cell growth media for improvement of standard bioprocess parameters, such as viable cell density, recombinant protein titer, and cell culture longevity. Since the hydrolysates were made from a crude material that currently has no specific purpose in animal cell technology, it seemed reasonable to screen for their potential benefit in that particular area [6]. Most of the reluctancy in using the hydrolysates originates from the lack of data about the oil cake composition variability as the result of the cultivar genetics and environmental conditions in which the plant was growing [29]. Methodologies of protein extraction and hydrolysis further contribute to this uncertainty.

### 3.1. Preparation of Hydrolysates

For preparation of the hydrolysates, three microbial proteases were used: *Alcalase, Neutrase,* and *Protamex.* Thus, the products of three independent enzymatic digestions

were appropriately marked as: FHA, FHN, and FHP. They were tested as CHO DP-12 cell culture media supplements with or without added fetal bovine serum (S). The hydrolysis of the proteins isolated from milled oil cake was carried out following the protocol described in our earlier work [19]. Optimal temperature and pH for each enzyme were maintained during the 4 h digestion process. The rates of hydrolysis were very fast, reaching a steady state after 15–20 min. By definition, the number of peptide bonds broken as a proportion of the total number of peptide bonds present is named the degree of hydrolysis (DH). Several techniques can be used to determine the DH of protein hydrolysates [30] and the one applied here is known as soluble nitrogen-TCA. It is an indirect method, but is very convenient and rapid. As shown in Figure 2A, similar DH was obtained with *Alcalase* (42.6%) and *Protamex* (41.7%), while *Neutrase* dissolved about one third of the peptide bonds of DH (33.9%). To visualize the efficiency of enzymatic digestion, we carried out a protein electrophoresis (SDS-PAGE) in reducing conditions (Figure 2B). The gel was loaded with samples containing about 50 μg of proteins. The electrophoretic profile of the lane with protein isolate (I) showed distribution of protein bands similar to those reported previously [17,31]. It had two strong bands at around 20 and 30 kDa and two weak bands at 10 and 45 kDa. The profile of the lanes with digested proteins (FHA, FHN, FHP) corresponds to the determined DH status, confirming that *Alcalase* performed the most thorough digestion with almost no visible bands in the gel. Two other enzymes were somewhat less efficient and produced significant amounts of peptides at about 10 kDa. A lane with FHN showed two protein fractions that were probably resistant to the enzyme. In our recent work on the antioxidant potential of flaxseed hydrolysates, we applied size-exclusion chromatography to evaluate the size and quantity of the fractions. The results showed that the low molecular peptides (<10 kDa) in FHA, FHN, and FHP were present at 95%, 91%, and 72%, respectively [19]. All three proteases that we used for hydrolysate production from flaxseed oil cake proteins had already been proven efficient in digestion of similar substrates. In most of these reports, however, the production of low molecular weight peptides (<10 kDa) did not follow the increase in DH, which can be explained by the differences in enzyme specificity [17,25,32].

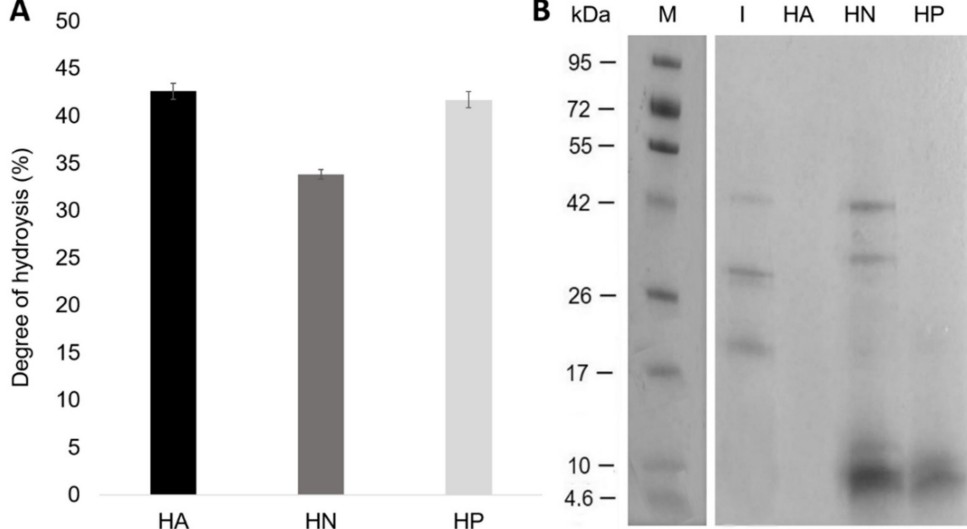

**Figure 2.** Three separate digestions of proteins isolated from flaxseed oil cake (I) were carried out using three different endopeptidases. (**A**) The produced hydrolysates (FHA, FHN, and FHP) had different degrees of hydrolysis. (**B**) The size of the peptide fractions in produced hydrolysates was evaluated by SDS-PAGE (M—markers).

*3.2. Adherent CHO Cell Culture in DMEM*

Adherent CHO cell culture in basal media with serum is still common in lab-scale research, especially in early biopharmaceutical development [33]. Here, we describe

the effects of DMEM supplemented with serum and flaxseed protein hydrolysates on adherent CHO DP-12 cells. For setting up our experiments, the cells were seeded in 24-well plates and grown in DMEM media with fetal bovine serum (S) and one of the three hydrolysates prepared using *Alcalase* (FHA), *Neutrase* (FHN), and *Protamex* (FHP). The usual portion of serum in cell culture media is 10% (*v/v*), which on average gives the total protein concentration of 5–6 g/L. To keep it simpler for the experiment design, the peptide concentration in our hydrolysates was set to 50 g/L. In order not to exceed the standard protein quantity, the media to be supplemented with hydrolysates contained only 5% (*v/v*) serum, while the tested hydrolysates were set at 1% and 5% (*v/v*). The formulations with less serum (2% and 1%) were also investigated, but they gave rather inconsistent results, and therefore are not presented here. Panel A in Figures 3–5 shows the growth curves of CHO DP-12 cells in the basal media supplemented with 5% serum and one of the three hydrolysates produced by *Alcalase* (FHA), *Neutrase* (FHN), and *Protamex* (FHP) at 1% or 5%. The control media formulation consisted of DMEM and serum at 5% and 10% (S5 and S10). The overall trend of the experiment was that the presence of hydrolysates in media with 5% serum promoted the cell growth. During the first half of the cultivation, the cells grew at a similar rate in all tested conditions. After day five, S5 and S10 seemed to slow down, reaching the cell density peak on day 7 or 8, with 250 to 320 $\times$ $10^3$ cells per well. The cultures containing hydrolysates reached their peak on day 8 or 9. The peaks were 20% to almost 90% higher than in the relevant controls. Moreover, it looks like the applied hydrolysate dose differently affected CHO DP-12 growth. In two cultures with FHA, the higher hydrolysate dose (5%) obtained the highest cell number, while in FHN and FHP cultures, the same effect was observed at the lower dose (1%). This might seem puzzling, with a typical dose–response curve in mind, but as discussed later, similar or even more confusing results were obtained in work with other peptide-based supplements [34]. Concerning the IgG production, we present it here as the relative volumetric productivity (Figures 3B, 4B and 5B). Such data comparison becomes more convenient when the cell line is not a high-yield producer and the cultivation is made in micro-scale systems with a fixed volume of media, as is often the case with recombinant protein production, so it was in our experiments, showing that the increase of IgG titer followed the rise in productive cell number. The significant productivity enhancement was found in the cultures with the best cell growth, i.e., among those that contained hydrolysates (5% FHA, 1% FHN, and 1% FHP). Both apparent improvements in cell performance were not entirely proportional, but from the metabolic point of view, it is often assumed the cell growth boost comes at the expense of reduced cell-specific production [35]. Looking at the glucose (Figures 3C, 4C and 5C) and lactate (Figures 3D, 4D and 5D) concentration profiles, we cannot conclude that the beneficial effects of the hydrolysates were exclusively of nutritional character. At the apparent decline phase of S5 and S10 cultures, the glucose was not entirely depleted, nor did the lactate reach a critical point, at least comparatively to the faster growing cultures with hydrolysates. There are reports with similar findings about wheat and soy peptide fractions in mouse hybridoma culture [9], or chickpea hydrolysates tested on THP-1 and Caco-2 cells [25]. In these studies, different fractions of one hydrolysate had different effects on hybridoma cell growth and productivity, or inversely, a single hydrolysate dose, had variable responses in different cell lines. Such inconsistent results suggest that hydrolysates may have a protective role, especially important in the late growth phase of adherent cultures, when the stress signals intensify because the growth surface becomes scarce and metabolic waste accumulates in excess. Our previous research confirms the proliferation stimulating effect of FHA, FHN, and FHP in HaCaT cells, but not in HeLa cells. In the same work, only FHA proved its antioxidant capacity, protecting the cells against induced oxidative stress [19]. Attributing these properties to increased amounts of low molecular weight peptides (<10 kDa) or to undefined leftover impurities can be speculated [9,19]. The best way to clarify this is to identify hydrolysate constituents, fraction them, and perform in vitro assessments.

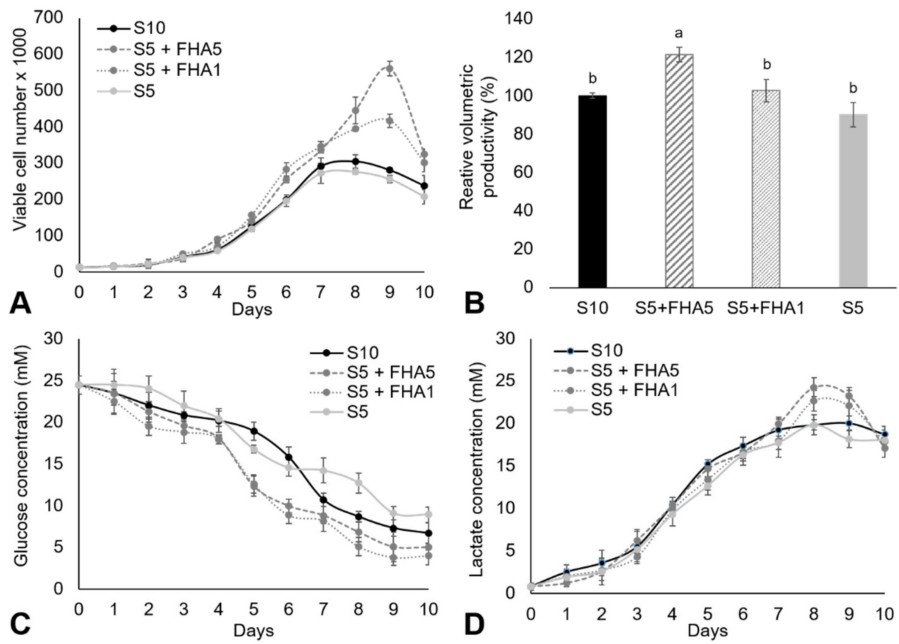

**Figure 3.** Flaxseed protein hydrolysate produced with *Alcalase* (FHA) was tested as the basal media supplement in CHO DP-12 cell culture. The hydrolysate was part of DMEM together with 5% serum (S5). The portions of FHA in the media were: 1% (S5 + FHA1) and 5% (S5 + FHA5). DMEM with 10% serum (S10) served as an additional control. (**A**) Growth curves of CHO DP-12 cells in different media formulations. (**B**) Relative volumetric productivity of CHO DP-12 cells in the media. Presented values followed by different lower-case letters (a–b) are significantly different from each other ($p < 0.05$), as measured by Tukey's HSD test. (**C**) Glucose concentration in the media. (**D**) Lactate concentration in the media.

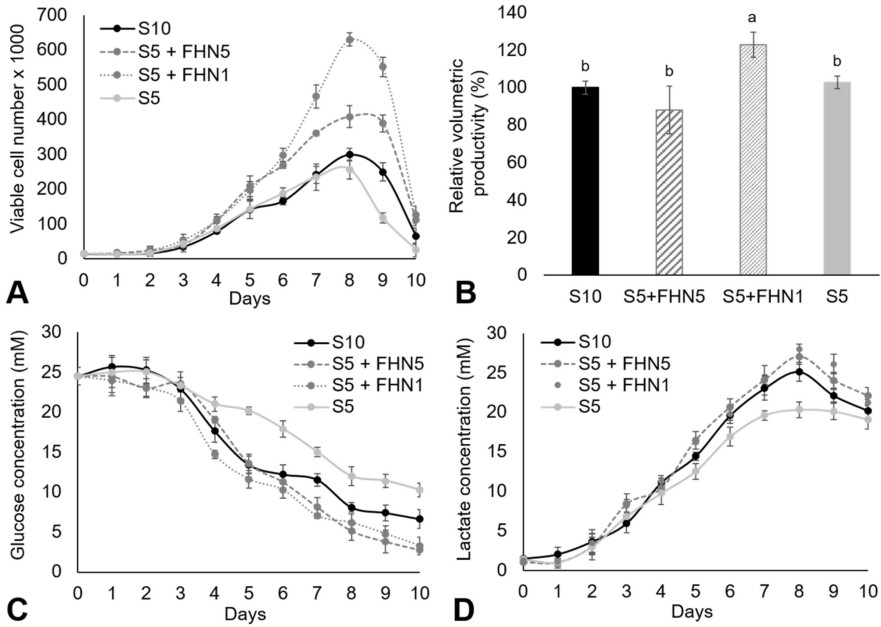

**Figure 4.** Flaxseed protein hydrolysate produced with *Neutrase* (FHN) was tested as the basal media supplement in CHO DP-12 cell culture. The hydrolysate was part of DMEM together with 5% serum (S5). The portions of FNA in the media were: 1% (S5 + FHN1) and 5% (S5 + FHN5). DMEM with 10% serum (S10) served as an additional control. (**A**) Growth curves of CHO DP-12 cells in different media formulations. (**B**) Relative volumetric productivity of CHO DP-12 cells in the media. Presented values followed by different lower-case letters (a–b) are significantly different from each other ($p < 0.05$), as measured by Tukey's HSD test. (**C**) Glucose concentration in the media. (**D**) Lactate concentration in the media.

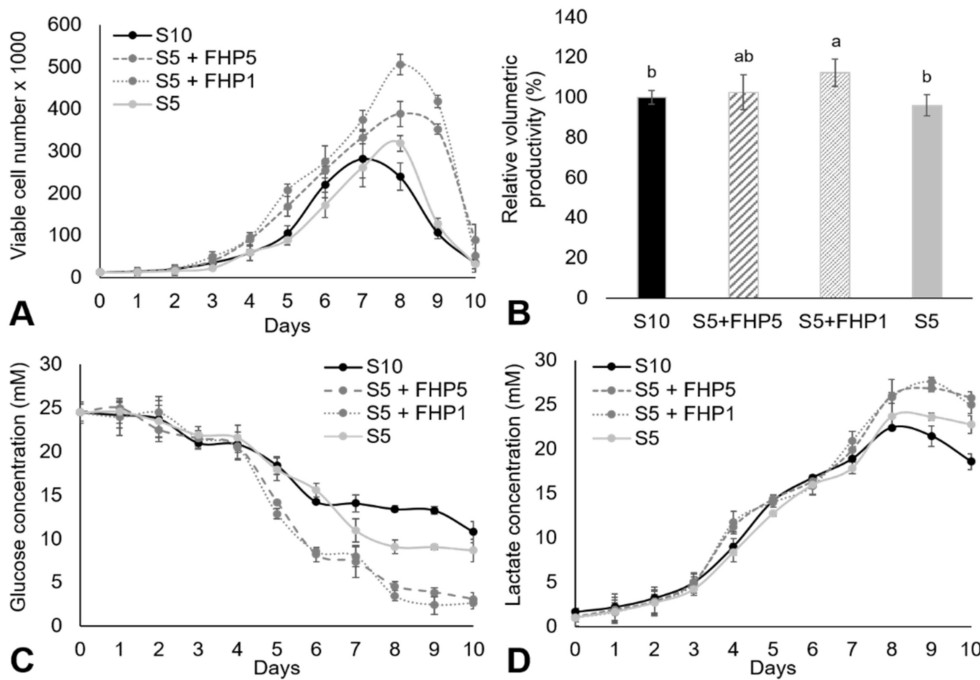

**Figure 5.** Flaxseed protein hydrolysate produced with *Protamex* (FHP) was tested as the basal media supplement in CHO DP-12 cell culture. The hydrolysate was part of DMEM together with 5% serum (S5). The portions of FNP in the media were: 1% (S5 + FHP1) and 5% (S5 + FHP5). DMEM with 10% serum (S10) served as an additional control. (**A**) Growth curves of CHO DP-12 cells in different media formulations. (**B**) Relative volumetric productivity of CHO DP-12 cells in the media. Presented values followed by different lower-case letters (a–b) are significantly different from each other ($p < 0.05$), as measured by Tukey's HSD test. (**C**) Glucose concentration in the media. (**D**) Lactate concentration in the media.

### 3.3. Suspension Cell Culture in CDM

The majority of manufacturing bioprocesses with mammalian cells are carried out with suspension cell cultures. Suspension systems are easy to scale and their engineering is well-understood, which enables quick implementation of bioprocessing protocols. Moreover, most of the suspension cell lines, including CHO, are adapted to proliferate in CDM. Depending on how specific and nutrient-rich they are, these media can be quite costly, so boosting them with protein hydrolysates can contribute to the reduction of bioprocessing expenses [21,36,37]. Such media supplementation barely hampers downstream processing, but it increases the product yield and it may affect the product glycosylation [16]. In this context, we tested in-house produced flaxseed protein hydrolysates as supplements in a commercial CDM. Prior to performing the experiments, adherent CHO DP-12 cells were adapted to suspension growth in the CDM, and as mentioned above, the new strain was named CHO DP-12-S. Here, we describe a simple preliminary assessment of a single hydrolysate concentration with this strain. Three media were prepared, each containing 1 g/L (2% *v/v*) of one of the hydrolysates. The choice of this particular concentration was made after the experiments with adherent cells, where it conveyed favorable results. However, with CHO DP-12-S, the addition of hydrolysates did not have any significant impact neither on viable cell number (Figure 6B) nor productivity (Figure 6A). The peaks of cell density were between 4.5 and $5 \times 10^6$ cells/mL, and there was less than a 10% difference in volumetric IgG yield. There were also no unexpected points in glucose consumption and lactate production curves (Figure 6C,D). Despite the rather neutral outcome, we plan to continue the investigation with suspension cells in CDM, particularly because the applied hydrolysate dose did not inhibit the cell growth. This implies that the failure to improve the suspension cell performance cannot be due to the presence of residual growth-inhibiting components, since five times higher hydrolysate concentration was used in the adherent

cell culture with serum. In our future experiments, the plan is to apply both lower and higher hydrolysate concentrations in order to observe positive effects on the cells, especially concerning the product titer. With a somewhat more comprehensive metabolic study, this will eventually allow us to speculate more competently about the benefits of CHO cell culture supplementation with flaxseed protein hydrolysate.

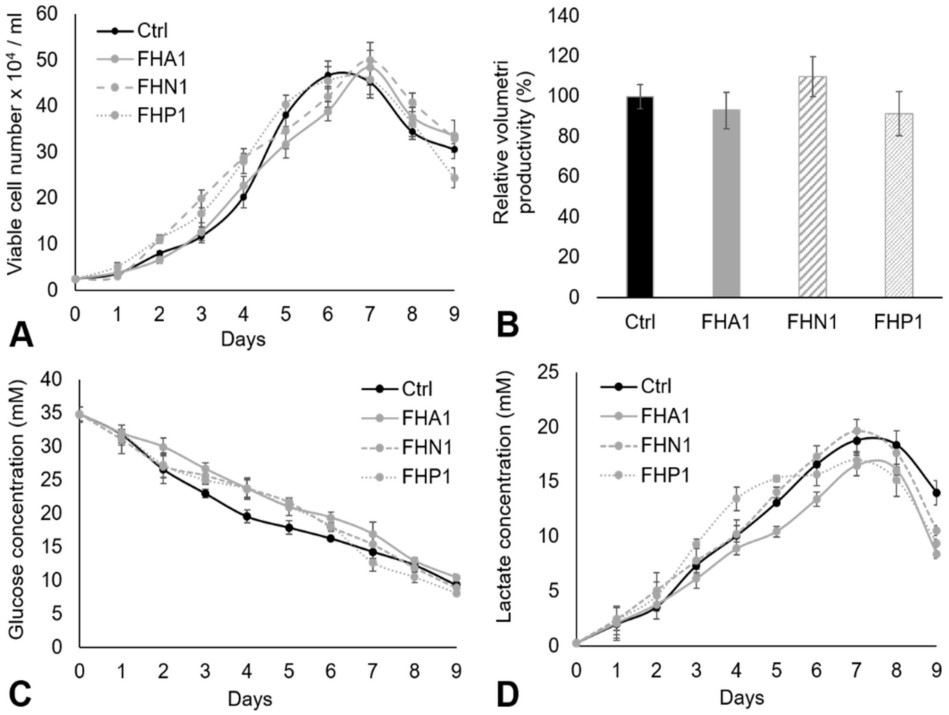

**Figure 6.** Flaxseed protein hydrolysates were part of CDM for investigation of their potentially favorable effects on CHO DP-12-S cells in suspension. The media formulations contained 1% (*v/v*) of one of the three hydrolysates, and were labeled FHA1, FHN1, and FHP1. The media without hydrolysate (Ctrl) served as a control. (**A**) Growth curves of CHO DP-12-S cells in different media formulations. (**B**) Relative volumetric productivity of CHO DP-12-S cells in the media. There was no significant difference in the product titer. (**C**) CDM glucose concentration. (**D**) Lactate concentration in the media.

## 4. Conclusions

Protein hydrolysates are complex mixtures of peptides, amino acids, and other low molecular weight compounds that are considered to be a suitable replacement for recombinant insulin and serum components in animal cell culture media. Although a shift towards CDM formulations has become standard, evaluation of hydrolysates for improvement of cell performance is still of interest in some technologies. In that respect, plant-derived hydrolysates are typically chosen over animal tissue hydrolysates due to safety concerns. In our work, we prepared flaxseed protein hydrolysates with three microbial proteases. Each proteolytic digestion delivered a hydrolysate of different DH and peptide fraction size. The most thorough digestion was obtained with *Alcalase,* slightly less efficient was *Protamex*, and the lowest DH was of *Neutrase*. The hydrolysates were first used to supplement DMEM with reduced amounts of serum and then tested on adherent IgG-producing CHO cells. Addition of the hydrolysates enhanced the cell growth, which consequently increased the final IgG titer. However, these improvements were not consistent with changes in hydrolysate doses applied. The cell culture had better growth and productivity figures with increased FHA portion in media, while it was the lower dose of FHN and FHP that provided advantageous results. Hydrolysate fractionation and peptide identification may define if such inconsistency is related to differences in protein digestion profiles. Additionally, it may help in understanding whether the role of the hydrolysates is more of a

protective or nutritive nature. In our final assessment, we screened for the favorable effects of the hydrolysates in suspension CHO culture with CDM. Media supplementation with a relatively low hydrolysate dose caused no obvious response regarding the culture growth, productivity, and longevity. This confirms the fact that commercial CDMs can be so meticulously formulated and rich in nutrients that adding small portions of supplements barely impacts cell performance. The obtained data is just a first screen of conditions and more fine-tuned optimizations could result in more convincing data, particularly concerning cell culture in CDM. No less important is the fact that this assessment proposes a new utility of flaxseed oil cake proteins, as a value-added by-product of food technology.

**Author Contributions:** Protein isolation, hydrolysate preparation, and cell cultivation, M.L. and S.B.; analytics, A.L.P. and K.R.; project leader, supervision, and text editing, V.G.S.; original draft preparation and supervision, I.S. All authors have read and agreed to the published version of the manuscript.

**Funding:** This research was funded by the Croatian Science Foundation (HRZZ), Grant No. IP-2016-06-3848.

**Data Availability Statement:** The data presented in this study are available upon request from the corresponding author.

**Conflicts of Interest:** The authors declare no conflict of interest.

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
