# Peer review of "Protein Hydrolysates from Flaxseed Oil Cake as a Media Supplement in CHO Cell Culture"

_resources, doi:10.3390/resources10060059_

Round 1

Reviewer 1 Report

The manuscript  "Protein hydrolysates from flaxseed oil cake as media supplement in CHO cell culture" presented for review contains elements of innovation, because waste material is used, which, according to the authors, has not been used for this purpose so far.
The flaxseed oil cake and its hydrolysis products (e.g. proteins or sugars) are the subject of many research, but the directions of their use are different. Therefore, the proposed experiments are scientifically relevant. The introduction very well introduces the purposefulness of the research, and the methodology has been very well described, allowing the presented experiences to be recreated. From the very beginning, the authors emphasize that the obtained results are not entirely satisfactory, but nevertheless bring some knowledge. Additionally, they should be continued to fully explain these mechanisms.
In my opinion, this work is an initial stage for further analyzes and conclusions.

Further analysis and research will be useful and should be published as a follow-up to the presented research.

I recommend the presented manuscript for publication in Resources, however, before this happens, Authors should verify some errors that are in the text.
Only some of them are listed below:
line 129 Lawry method?
line 241 "specificity. [13,21,28]," - redundant periods and commas e.t.c.
I am also asking you to correct the inaccuracies between the lines:
143 Each loaded sample contained 50 μg of peptides
and 225-226 The gel was loaded with samples containing about 5 μg of proteins.
You should also review the list of literature again and adjust the selected items to the pattern.

Author Response

Dear Reviewer,

Please find the authors' responses in the attachment.

Kind regards.

Reviewer 2 Report

This manuscript was discussed the re-use of flaxseed oil cake and apply to as media of CHO cell culture. The aim of this manuscript was good but lack of novelty. I still have some comments as below:

  1. Why author choice CHO cell, because if the author change another cell, the same reuslts could also found?
  2. the abstract in the 18-21, the author should delete this sentence.
  3. In introduction, in line 27-38, this is the basic information for reader, the author should delete this paragraph, or re-write in another introduction topic, for example: flaxseed oil cake, CHO cell, the recent research related to the reusement of oil cake.
  4. In line 65-66, the fig. 1 should move to the material and method, and author should give more detail description about the advantage of add protein hydrolysates to the median, because we can choice other sources of protein hydrolysates?
  5. the main composition of flaseed protein isolate and protein hydrolysates should be provided.
  6. In Fig. 2, the author sholud also provide the standard of protein molecule for SDS-PAGE.
  7. the residual nitrogen content for Fig.5 should be provide ?
  8. the author did not provide the results of IgG?

Author Response

(The authors gave the same response as above.)

Reviewer 3 Report

The authors mentioned that the amount of proteins in flaxseed oil is in between 20 to 30 %. Then, the importance of some amino acids with respect to free radical scavenging activity and linoletic acid oxidation inhibition; these amino acids are arginine, lysine, valine and leucine. There are some questions which are interesting and could be clarified:

  1. Do the authors know the amino acid composition of flaxseed oil ?
  2. What about amino acid composition of 3 types of protein hydrolysates prepared in the study?
  3. It would be useful for the readers if they can compare arginine, lysine, valine and leucine content in flaxseed oil hydrolysates in contrast with other types of (plant-derived protein) hydrolysates.

Page 2, line 71: A comma is missing between “lysine” and “leucine”.

Author Response

(The authors gave the same response as above.)

Reviewer 4 Report

This is a very interesting study about flaxseed protein hydrolysates applied as media supplements in CHO cell culture.

The manuscript is well written and it highlights important scientific point. The abstract presents very clear the objectives of the study. The results can be useful for research in the field, even if the obtained data is just a first screen of concerning cell culture in CDM.

However, there are some minor observations:

In my opinion, it would seem appropriate to explain in one sentence, in the Introduction section, what CHO cells means and what are the advantages of using them.

Line 382. I suggest you give up the quote [35] and reformulate this sentence. Only your personal conclusions should appear in the conclusions section, not those of other authors. Eventually, you can use that quote in the Discussion section.

Author Response

(The authors gave the same response as above.)

Round 2

Reviewer 2 Report

The manuscript had corrected according to my comments.